# Generalizing Goal-Conditioned Reinforcement Learning with Variational Causal Reasoning

**Wenhao Ding**[1], **Haohong Lin**[1], **Bo Li**[2], **Ding Zhao**[1]
[1]Carnegie Mellon University
[2]University of Illinois Urbana-Champaign
{wenhaod, haohongl}@andrew.cmu.edu, lbo@illinois.edu, dingzhao@cmu.edu

## Abstract

As a pivotal component to attaining generalizable solutions in human intelligence, reasoning provides great potential for reinforcement learning (RL) agents' generalization towards varied goals by summarizing part-to-whole arguments and discovering cause-and-effect relations. However, how to discover and represent causalities remains a huge gap that hinders the development of causal RL. In this paper, we augment Goal-Conditioned RL (GCRL) with *Causal Graph (CG)*, a structure built upon the relation between objects and events. We novelly formulate the GCRL problem into variational likelihood maximization with CG as latent variables. To optimize the derived objective, we propose a framework with theoretical performance guarantees that alternates between two steps: using interventional data to estimate the posterior of CG; using CG to learn generalizable models and interpretable policies. Due to the lack of public benchmarks that verify generalization capability under reasoning, we design nine tasks and then empirically show the effectiveness of the proposed method against five baselines on these tasks. Further theoretical analysis shows that our performance improvement is attributed to the virtuous cycle of causal discovery, transition modeling, and policy training, which aligns with the experimental evidence in extensive ablation studies. Code is available on https://github.com/GilgameshD/GRADER.

## 1 Introduction

The generalizability, which enables an algorithm to handle unseen tasks, is fruitful yet challenging in multifarious decision-making domains. Recent literature [1, 2, 3] reveals the critical role of reasoning in improving the generalization of reinforcement learning (RL). However, most off-the-shelf RL algorithms [4] have not regarded reasoning as an indispensable accessory, thus usually suffering from data inefficiency and performance degradation due to the mismatch between training and testing settings. To attain generalization at the testing stage, some efforts were put into incorporating domain knowledge to learn structured information, including sub-task decomposition [5] and program generation [6, 7, 8, 9, 10], which guide the model to solve complicated tasks in an explainable way. However, such symbolism-dominant methods heavily depend on the re-usability of sub-tasks and pre-defined grammars, which may not always be accessible in decision-making tasks.

Inspired by the close link between reasoning and the cause-and-effect relationship, causality is recently incorporated to compactly represent the aforementioned structured knowledge in RL training [11]. Based on the form of causal knowledge, we divide the related works into two categories, i.e., *implicit* and *explicit* causation. With *implicit* causal representation, researchers ignore the detailed causal structure. For instance, [12] extracts invariant features as one node that influences the reward function, while the other node consists of task-irrelevant features [13, 14, 15, 16]. This neat structure has good scalability but requires access to multiple environments that share the

36th Conference on Neural Information Processing Systems (NeurIPS 2022).

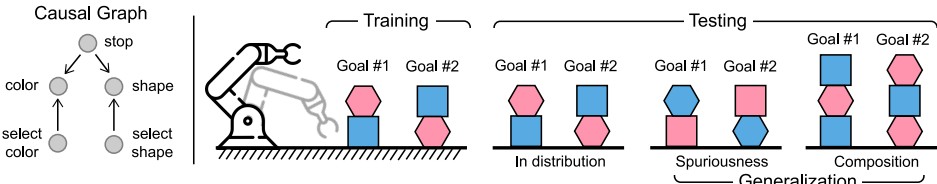

Figure 1: The robot picks and places objects to achieve given goals. **Left:** The causal graph of this task. **Right:** Training setting: the hexagon is always pink and the box is always blue. Three testing settings: (1) In distribution: the same as the training setting. (2) Spuriousness: swap the color and shape to break the spurious correlation. (3) Composition: increase the height of the goal.

same invariant feature [12, 16, 17]. In contrast, one can turn to the *explicit* side by estimating detailed causal structures [18, 19, 20, 21], which uses directed graphical models to capture the causality in the environment. A pre-request for this estimation is the object-level or event-level abstraction of the observation, which is available in most tasks and also becoming a frequently studied problem [22, 23, 24]. However, existing *explicit* causal reasoning RL models either require the true causal graph [25] or rely on heuristic design without theoretical guarantees [18].

In this paper, we propose *GeneRAlizing by DiscovERing (GRADER)*, a causal reasoning method that augments the RL algorithm with data efficiency, interpretability, and generalizability. We mainly focus on Goal-Conditioned RL (GCRL) [26], where different goal distributions during training and testing reflect the generalization. We formulate the GCRL into a probabilistic inference problem [27] with a learnable causal graph as the latent variable. This novel formulation naturally explains the learning objective with three components – transition model learning, planning, and causal graph discovery – leading to an optimization framework that alternates between causal discovery and policy learning to gain generalizability. Under some mild conditions, we prove the unique identifiability of the causal graph and the theoretical performance guarantee of the proposed framework.

To demonstrate the effectiveness of the proposed method, we conduct comprehensive experiments in environments that require strong reasoning capability. Specifically, we design two types of generalization settings, i.e., spuriousness and composition, and provide an example to illustrate these settings in Figure 1. The evaluation results confirm the advantages of our method in two aspects. First, the proposed data-efficient discovery method provides an explainable causal graph yet requires much fewer data than previous methods, increasing data efficiency and interpretability during task solving. Second, simultaneously discovering the causal graph during policy learning dramatically increases the success rate of solving tasks. In summary, the contribution of this paper is threefold:

- We use the causal graph as a latent variable to reformulate the GCRL problem and then derive an iterative training framework from solving this problem.

- We prove that our method uniquely identifies true causal graphs, and the performance of the iterative optimization is guaranteed with a lower bound given converged transition dynamics.

- We design nine tasks in three environments that require strong reasoning capability and show the effectiveness of the proposed method against strong baselines on these tasks.

## 2 Problem Formulation and Preliminary

We start by discussing the setting we consider in this paper and the assumptions required in causal reasoning. Then we briefly introduce the necessary concepts related to causality and causal discovery.

### 2.1 Factorized Goal-conditioned RL

We assume the environment follows the Goal-conditioned Markov Decision Process (MDP) setting with full observation. This setting is represented by a tuple $\mathcal{M} = (\mathcal{S}, \mathcal{A}, \mathcal{P}, \mathcal{R}, G)$, where $\mathcal{S}$ is the state space, $\mathcal{A}$ is the action space, $\mathcal{P}$ is the probabilistic transition model, $G \subset \mathcal{S}$ is the goal space which is a set of assignment of values to states, and $r(s, g) = \mathbb{1}(s = g) \in \mathcal{R}$ is the sparse deterministic reward function that returns 1 only if the state $s$ match the goal $g$. In this paper, we focus on the goal-conditioned generalization problem, where the goal for training and testing stages will be sampled from different distributions $p_{\text{train}}(g)$ and $p_{\text{test}}(g)$. We refer to a goal $g \in G$ as a task

and use these two terms interchangeably. To accomplish the causal discovery methods, we make a further assumption similar to [20, 28] for the state and action space:

**Assumption 1** (Space Factorization). *The state space $\mathcal{S} = \{\mathcal{S}_1 \times \cdots \times \mathcal{S}_M\}$ and action space $\mathcal{A} = \{\mathcal{A}_1 \times \cdots \times \mathcal{A}_N\}$ can be factorized to disjoint components $\{\mathcal{S}_i\}_{i=1}^M$ and $\{\mathcal{A}_i\}_{i=1}^N$.*

The components representing one event or object's property usually have explicit semantic meanings for better interpretability. This assumption can be satisfied by state and action abstraction, which has been widely investigated in [22, 23, 24]. Such factorization also helps deal with the high-dimensional states since it could be intractable to treat each dimension as one random variable [18].

## 2.2 Causal Reasoning with Graphical Models

Reasoning with causality relies on specific causal structures, which are commonly represented as directed acyclic graphs (DAGs) [29] over variables. Consider random variables $\boldsymbol{X} = (X_1, \ldots, X_d)$ with index set $\boldsymbol{V} := \{1, \ldots, d\}$. A graph $\mathcal{G} = (\boldsymbol{V}, \mathcal{E})$ consists of nodes $\boldsymbol{V}$ and edges $\mathcal{E} \subseteq \boldsymbol{V}^2$ with $(i, j)$ for any $i, j \in \boldsymbol{V}$. A node $i$ is called a parent of $j$ if $e_{ij} \in \mathcal{E}$ and $e_{ji} \notin \mathcal{E}$. The set of parents of $j$ is denoted by $\boldsymbol{PA}_j^{\mathcal{G}}$. We formally discuss the graph representation of causality with two definitions:

**Definition 1** (Structural Causal Models [29]). *A structural causal model (SCM) $\mathfrak{C} := (\boldsymbol{S}, \boldsymbol{U})$ consists of a collection $\boldsymbol{S}$ of $d$ functions $X_j := f_j(\boldsymbol{PA}_j^{\mathcal{G}}, U_j)$, $j \in [d]$, where $\boldsymbol{PA}_j \subset \{X_1, \ldots, X_d\} \backslash \{X_j\}$ are called parents of $X_j$; and a joint distribution $\boldsymbol{U} = \{U_1, \ldots, U_d\}$ over the noise variables, which are required to be jointly independent.*

**Definition 2** (Causal Graph [29], CG). *The causal graph $\mathcal{G}$ of an SCM is obtained by creating one node for each $X_j$ and drawing directed edges from each parent in $\boldsymbol{PA}_j^{\mathcal{G}}$ to $X_j$.*

We note that CG describes the structure of the causality, and SCM further considers the specific causation from the parents of $X_j$ to $X_j$ via $f_j$ as well as exogenous noises $U_j$. To uncover the causal structure from data distribution, we assume that the CG satisfies the *Markov Property* and *Faithfulness* [29], which make the independences consistent between the joint distribution $P(X_1, \ldots, X_n)$ and the graph $\mathcal{G}$. We also follow the *Causal Sufficiency* assumption [30] that supposes we have measured all the common causes of the measured variables.

Existing work [31, 20] believes that two objects have causality only if they are close enough while there is no edge between them if the distance is large. Instead of using such a local view of the causality, we assume the causal graph is consistent across all time steps, which also handles the local causal influence. The specific influence indicated by edges is estimated by the function $f_j(\boldsymbol{PA}_j, U_j)$.

# 3 Generalizing by Discovering (GRADER)

With proper definitions and assumptions, we now look at the proposed method. We first derive the framework of GRADER by formulating the GCRL problem as a latent variable model, which provides a variational lower bound to optimize. Then, we divide this objective function into three parts and iteratively update them with a performance guarantee.

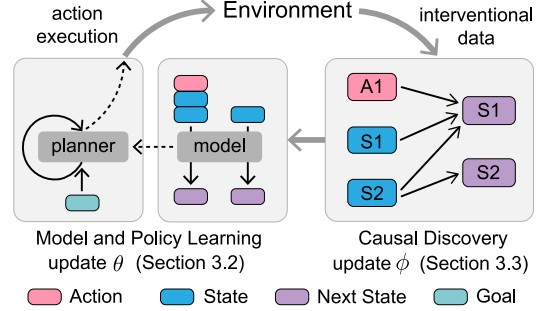

Figure 2: The paradigm of GRADER.

## 3.1 GCRL as Latent Variable Models

The general objective of RL is to maximize the expected reward function w.r.t. a learnable policy model $\pi$. Particularly, in the goal-conditioned setting, such objective is represented as $\max_\pi \mathbb{E}_{\tau \sim \pi, g \sim p(g)}[\sum_{t=0}^T r(s^t, g)]$, where $p(g)$ is the distribution of goal and $\tau := \{s^0, a^0, \ldots, s^T\}$ is the action-state trajectory with maximal time step $T$. The trajectory ends only if the goal is achieved $g = s^T$ or the maximal step is reached.

Inspired by viewing the RL problem as probabilistic inference [32, 27], we replace the objective from *"How to find actions to achieve the goal?"* to *"what are the actions if we achieve the goal?"*, leading

to a likelihood maximization problem for $p(\tau|s^*)$ with $s^* := \mathbb{1}(g = s^T)$. Different from previous work [33] that recasts actions as latent variables and infers actions that result in "observed" high reward, we decompose $p(\tau|s^*)$ with $\mathcal{G}$ as the latent variable to get the evidence lower bound (ELBO)

$$\log p(\tau|s^*) = \log \int p(\tau|\mathcal{G}, s^*)p(\mathcal{G}|s^*)d\mathcal{G} \geq \mathbb{E}_{q(\mathcal{G}|\tau)}[\log p(\tau|\mathcal{G}, s^*)] - \mathbb{D}_{\mathrm{KL}}[q(\mathcal{G}|\tau)||p(\mathcal{G})], \quad (1)$$

where the prior $p(\mathcal{G})$ and variational posterior $q(\mathcal{G}|\tau)$ represent distributions over graph structures, i.e, the probability of the existences of edges in graphs. $\mathbb{D}_{\mathrm{KL}}$ denotes the Kullback–Leibler (KL) divergence between two graphs, which will be explained in Section 3.3. Recall that the space of goal $G$ is a subset of the state, we extend the meaning of $g$ and assume all trajectories achieve the goal in the final state [34], i.e., $g = s^T$. Such extension makes it possible to further decompose the first term of (1) as (refer to Appendix A.1.2)

$$\log p(\tau|\mathcal{G}, s^*) = \log p(s^0) + \sum_{t=0}^{T-1} \log p(s^{t+1}|s^t, a^t, \mathcal{G}) + \sum_{t=0}^{T-1} \log \pi(a^t|s^t, s^*, \mathcal{G}) + \log p(g). \quad (2)$$

Here, we use the fact that $\mathcal{G}$ is effective in both the transition model $p(s^{t+1}|a^t, s^t, \mathcal{G})$ and the policy model $\log p(a^t|s^t, s^*, \mathcal{G})$, $g$ only influences the policy model, and the initial state $s_0$ depends neither on $\mathcal{G}$ nor $g$. We also assume that both initial state $\log p(s_0)$ and goal $\log p(g)$ follow the uniform distribution. Thus, the first and last terms of (2) are constants. The policy term $\pi$, selecting action $a^t$ according to both current state $s^t$ and goal $g$, is implemented with the planning method and is further discussed in Section 3.2. Finally, we maximize the likelihood $p(\tau|s^*)$ with the following reformulated ELBO as the objective

$$\mathcal{J}(\theta, \phi) = \mathbb{E}_{q_\phi(\mathcal{G}|\tau)} \sum_{t=0}^{T-1} \left[ \log p_\theta(s^{t+1}|s^t, a^t, \mathcal{G}) + \log \pi_\theta(a^t|s^t, s^*, \mathcal{G}) \right] - \mathbb{D}_{\mathrm{KL}}[q_\phi(\mathcal{G}|\tau)||p(\mathcal{G})] \quad (3)$$

where $\theta$ is the shared parameter of transition model $p_\theta(s^{t+1}|a^t, s^t, \mathcal{G})$ and policy $\pi_\theta(a^t|s^t, s^*, \mathcal{G})$, and $\phi$ is the parameter of causal graph $q_\phi(\mathcal{G}|\tau)$. To efficiently solve this optimization problem, we iteratively updates parameter $\phi$ (causal discovery, Section 3.3) and parameter $\theta$ (model and policy learning, Section 3.2), as shown in Figure 2. Intuitively, these processes can be respectively viewed as the discovery of graph and the update of $f_i$, which share tight connections as discussed in Section 2.2.

## 3.2 Model and Policy Learning

Let us start with a simple case where we already obtain a $\mathcal{G}$ and use it to guide the learning of parameter $\theta$ via $\max_\theta \mathcal{J}(\theta, \phi)$. Since the KL divergence of $\mathcal{J}(\theta, \phi)$ does not involve $\theta$, we only need to deal with the first expectation term, i.e., the likelihood of transition model and policy. For the transition $p_\theta(s^{t+1}|a^t, s^t, \mathcal{G})$, we connect it with causal structure by further defining a particular type of CG and denote it as $\mathcal{G}$ in the rest of this paper:

**Definition 3** (Transition Causal Graph). *We define a bipartite graph $\mathcal{G}$, whose vertices are divided into two disjoint sets $\mathcal{U} = \{\mathcal{A}^t, \mathcal{S}^t\}$ and $\mathcal{V} = \{\mathcal{S}^{t+1}\}$. $\mathcal{A}^t$ represents action nodes at step $t$, $\mathcal{S}^t$ state nodes at step $t$, and $\mathcal{S}^{t+1}$ the state nodes at step $t + 1$. All edges start from set $\mathcal{U}$ and end in set $\mathcal{V}$.*

**Model learning.** This definition builds the causal graph between two consecutive time steps, which indicates that the values of states in step $t + 1$ depend on values in step $t$. It also implies that the interventions [29] on nodes in $\mathcal{U}$ are directly obtained since they have no parent nodes. We denote the marginal distribution of $\mathcal{S}$ as $p_{\mathcal{I}_\pi^s}$, which is collected by RL policy $\pi$. Combined with the Definition 1 of SCM, we find that $p_\theta(s^{t+1}|a^t, s^t, \mathcal{G})$ essentially approximates a collection of functions $f_j$ following the structure $\mathcal{G}$, which take as input the values of parents of the state node $s_j$ and outputs the value $s_j$. Thus, we propose to model the transition corresponding to $\mathcal{G}$ with a collection of neural networks $f_\theta(\mathcal{G}) := \{f_{\theta_j}\}_{j=1}^M$ to obtain

$$s_j^{t+1} = f_{\theta_j}([\mathbf{PA}_j^{\mathcal{G}}]^t, U_j), \quad (4)$$

where $[\mathbf{PA}_j^{\mathcal{G}}]^t$ represents the values of all parents of node $s_j^t$ at time step $t$ and $U_j$ follows Gaussian noise $U_j \sim \mathcal{N}(0, \mathbf{I})$. In practice, we use Gated Recurrent Unit [35] as $f_j$ because it supports varying numbers of input nodes. We take $s_j^t$ as the initial hidden embedding and the rest parents $[\mathbf{PA}_j^{\mathcal{G}} \backslash s_j]^t$

as the input sequence to $f_j$. The entire model is optimized by stochastic gradient descent with the log-likelihood $\log p_\theta(s^{t+1}|a^t, s^t, \mathcal{G})$ as objective.

**Policy learning with planning.** Then we turn to the policy term $\pi_\theta(a^t|s^t, s^*, \mathcal{G})$ in $\mathcal{J}(\theta, \phi)$. We optimize it with planning methods that leverage the estimated transition model. Specifically, the policy aims to optimize an action-state value function $Q(s^t, a^t) = \mathbb{E}\left[\sum_{t'=0}^{H} \gamma^{t'} r(s^{t'+t}, a^{t'+t})|s^t, a^t\right]$, which can be obtained by unrolling the transition model with a horizon of $H$ steps and discount factor $\gamma$. In practice, we use model predictive control (MPC) [36] with random shooting [37], which selects the first action in the fixed-horizon trajectory that has the highest action-state value $Q(s^t, a^t)$, i.e. $\hat{\pi}(s^t) = \arg\max_{a^t \in \mathcal{A}} Q_\theta^\mathcal{G}(s^t, a^t)$. The formulation we derived so far is highly correlated to the model-based RL framework [38]. However, the main difference is that we obtain it with variational inference by regarding the causal graph as a latent variable.

### 3.3 Data-Efficient Causal Discovery

In this step, we relax the assumption of knowing $\mathcal{G}$ and aim to estimate the posterior distribution $q(\mathcal{G}|\tau)$ to optimize ELBO (3) w.r.t. parameter $\phi$. In most score-based methods [39], likelihood is used to evaluate the correctness of the causal graph, i.e., a better causal graph leads to a higher likelihood. Since the first term of (3) represents the likelihood of the transition model, we convert the problem of $\max_\phi \mathcal{J}(\theta, \phi)$ to the causal discovery that finds the true causal graph based on collected data samples. As for the second term of (3), the following proposition shows that the KL divergence between $q_\phi(\mathcal{G}|\tau)$ and $p(\mathcal{G})$ can be approximated by a sparsity regularization (proof in Appendix A.4.2).

**Proposition 1** (KL Divergence as Sparsity Regularization). *With entry-wise independent Bernoulli prior $p(\mathcal{G})$ and point mass variational distribution $q(\mathcal{G}|\tau)$ of DAGs, $\mathbb{D}_{KL}[q_\phi\|p]$ is equivalent to an $\ell_1$ sparsity regularization for the discovered causal graph.*

---

**Algorithm 1:** GRADER Training

**Input:** Trajectory buffer $\mathcal{B}_\tau$, Causal graph $\mathcal{G}$, Transition model $f_\theta$, causal discovery threshold $\eta$

**while** $\theta$ *not converged* **do**
  // Policy from planning
  Sample a goal $g \sim p_{\text{train}}(g)$
  **while** $t < T$ **do**
    $a^t \leftarrow \text{Planner}(f_\theta, s^t, g)$
    $s^{t+1}, r^t \leftarrow \text{Env}(a^t, g)$
    $\mathcal{B}_\tau \leftarrow \mathcal{B}_\tau \cup \{a^t, s^t, s^{t+1}\}$
  // Estimate causal graph
  **for** $i \leq M + N$ **do**
    **for** $j \leq M$ **do**
      Infer edge $e_{ij} \leftarrow q_\phi(\cdot|\mathcal{B}, \eta)$
  // Learn transition model
  Update $f_\theta(\mathcal{G})$ via (4) with $\mathcal{B}$

---

We restrict the posterior $q_\phi(\mathcal{G}|\tau)$ to point mass distribution and use a threshold $\eta$ to control the sparsity. We perform the discovery process from the classification perspective by proposing binary classifiers $q_\phi(e_{ij}|\tau, \eta)$ to determine the existence of an edge $e_{ij}$. This classifier $q_\phi(e_{ij}|\tau, \eta)$ is implemented by statistic *Independent Test* [40] and $\eta$ is the threshold for the p-value of the hypothesis. A larger $\eta$ corresponds to harder sparsity constraints, leading to a sparse $\mathcal{G}$ since two nodes are more likely to be considered independent. According to the definition 3, we only need to conduct classification to edges connecting nodes between $\mathcal{U}$ and $\mathcal{V}$. If two nodes are dependent, we add one edge directed from the node in $\mathcal{U}$ to the node in $\mathcal{V}$. This definition also ensures that we always have $q(\mathcal{G}|\tau) \in \mathcal{Q}_{DAG}$, where $\mathcal{Q}_{DAG}$ is the class of DAG. With this procedure, we identify a unique CG $\mathcal{G}^*$ under optimality:

**Proposition 2** (Identifiability). *Given an oracle independent test, with an optimal interventional data distribution $p^*_{\mathcal{I}^s_\pi}$, causal discovery obtains $\phi^*$ that correctly tells the independence between any two nodes, then the causal graph is uniquely identifiable, with $e^*_{ij} = q_{\phi^*}(e_{ij}|\tau), \forall i \in [M + N], j \in [M]$.*

In practice, we use $\chi^2$-test for discrete variables and the Fast Conditional Independent Test [40] for continuous variables. The sample size needed for obtaining the oracle test has been widely investigated [41]. However, testing with finite data is not a trivial problem, as stated in [42], especially when the data is sampled from a Goal-conditioned MDP. Usually, the random policy is not enough to satisfy the oracle assumption because some nodes cannot be fully explored when the task is complicated and has a long horizon. To make this assumption empirically possible, it is necessary to simultaneously optimize $\pi_\theta(a^t|s^t, s^*, \mathcal{G})$ to access more samples close to finishing the task, which is further analyzed in Section 3.4. We also empirically support this argument in Section 4.2 and provide a detailed theoretical proof in Appendix A.3 and A.4.

## 3.4 Analysis of Performance Guarantee

The entire pipeline of GRADER is summarized in Algorithm 1. To analyze the performance of the optimization of (3), we first list important lemmas that connect previous steps and then show that the iteration of these steps in *GRADER* leads to a virtuous cycle.

By the following lemmas, we show the following performance guarantees step by step. Lemma 1 shows model learning is monotonically better at convergence given a better causal graph from causal discovery. Then the learned transition model helps to improve the lower bound of the value function during planning according to Lemma 2. Lemma 3 reveals the connection between policy learning and interventional data distribution, which in turn improves the quality of our causal discovery, as is shown in Lemma 4 and Proposition 2.

**Lemma 1** (Monotonicity of Transition Likelihood). *Assume $\mathcal{G}^* = (V, E^*)$ be the true CG, for two CG $\mathcal{G}_1 = (V, E_1)$ and $\mathcal{G}_2 = (V, E_2)$, if $SHD(\mathcal{G}_1, \mathcal{G}^*) < SHD(\mathcal{G}_2, \mathcal{G}^*)$, $\exists\, e$, s.t. $E_1 \cup \{e\} = E_2$, when transition model $\theta$ converges, the following inequality holds for the transition model in (3):*

$$\log p_\theta(s^{t+1}|a^t, s^t, \mathcal{G}^*) \geq \log p_\theta(s^{t+1}|a^t, s^t, \mathcal{G}_1) \geq \log p_\theta(s^{t+1}|a^t, s^t, \mathcal{G}_2) \quad (5)$$

*where SHD is the Structural Hamming Distance defined in Appendix A.2.*

**Lemma 2** (Bounded Value Function in Policy Learning). *Given a planning horizon $H \to \infty$, if we already have an approximate transition model $\mathbb{D}_{TV}(\hat{p}(s'|s, a), p(s'|s, a)) \leq \epsilon_m$, the approximate policy $\hat{\pi}$ achieves a near-optimal value function (refer to Appendix A.3 for detailed analysis):*

$$\|V^{\pi^*}(s) - V^{\hat{\pi}}(s)\|_\infty \leq \frac{\gamma}{(1-\gamma)^2}\epsilon_m \quad (6)$$

**Lemma 3** (Policy Learning Improves Interventional Data Distribution). *With a step reward $r(s, a) = \mathbb{1}_{\partial}\mathbb{1}(s = g)$, we show that the value function determines an upper bound for TV divergence between the interventional distribution with its optimal (proof details in Appendix A.3):*

$$\mathbb{D}_{TV}(p_{\mathcal{I}^s_\pi}, p_g) \leq 1 - (1-\gamma)V^\pi(s). \quad (7)$$

*where $p_{\mathcal{I}^s_\pi}$ is the marginal state distribution in interventional data, and $p_g$ is the goal distribution. A better policy with larger $V^\pi(s)$ enforces the distribution of interventional data toward the goal.*

**Lemma 4** (Interventional Data Benefits Causal Discovery). *For $\epsilon_g = \min_{p_g > 0} p_g$, $\mathbb{D}_{TV}(p_{\mathcal{I}^s_\pi}, p_g) < \epsilon_g$, the error of our causal discovery is upper bounded with $\mathbb{E}_{\hat{\mathcal{G}}}[SHD(\hat{\mathcal{G}}, \mathcal{G}^*)] \leq |\mathcal{S}| - 1$.*

After the close-loop analysis of our model, we are now able to analyze the overall performance of the proposed framework. Under the construction of $p_\theta(s^{t+1}|a^t, s^t, \mathcal{G})$ with NN-parameterized functions, the following theorem shows that the learning process will guarantee to perform a close estimation of true ELBO under the iterative optimization among model learning, planning, and causal discovery.

**Theorem 1.** *With T-step approximate transition dynamics $\mathbb{D}_{TV}\left(\hat{p}(s'|s, a), p(s'|s, a)\right) \leq \epsilon_m$, if the goal distribution satisfies $\epsilon_g > \frac{\gamma}{1-\gamma}\epsilon_m$, and the distribution prior CG is entry-wise independent Bernoulli($\epsilon_\mathcal{G}$), GRADER guarantees to achieve an approximate ELBO $\hat{\mathcal{J}}$ with the true ELBO $\mathcal{J}^*$:*

$$\|\mathcal{J}^*(\theta, \phi) - \hat{\mathcal{J}}(\hat{\theta}, \hat{\phi})\|_\infty \leq \left[1 + \frac{\gamma}{(1-\gamma)^2}\right]\epsilon_m T + \log\left(\frac{1-\epsilon_\mathcal{G}}{\epsilon_\mathcal{G}}\right)(|\mathcal{S}| - 1), \quad (8)$$

An intuitive understanding of the performance guarantee is that a better transition model indicates a better approximation of objective $\mathcal{J}$. The proof of this theorem and corresponding empirical results are in Appendix A.5.

## 4 Experiments

In this section, we first discuss the setting of our designed environments as well as the baselines used in the experiments. Then, we provide the numerical results and detailed discussions to answer the following important research questions: **Q1.** Compared to existing strong baselines, how does GRADER gain performance improvement under both in-distribution and generalization settings? **Q2.**

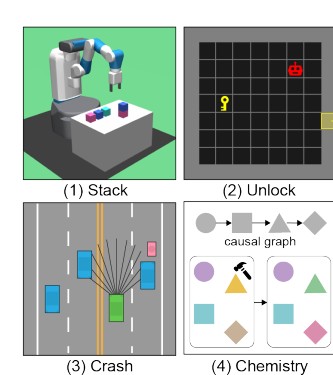

(1) Stack    (2) Unlock

(3) Crash    (4) Chemistry

Compared to an offline random policy, how does a well-trained policy improve the results of causal discovery? **Q3.** Compared to score-based causal discovery, does the proposed data-efficient causal discovery pipeline guarantee identifying the true causal graph as stated in Section 3.3? **Q4.** Considering the correctness of causal graphs, how does the imperfect causal graph influence the task-solving performance of GCRL agents?

## 4.1 Environments and Baselines

Since most commonly used RL benchmarks do not explicitly require causal reasoning for generalization, we design three new environments, which are shown in Figure 3 (excluding *Chemistry* [43]). These environments use the true state as observation to disentangle the reasoning task from visual understanding. For each environment, we design three settings – in-distribution (*I*), spuriousness (*S*), and composition (*C*) – corresponding to different goal distributions for generalization. We use $p_{\text{train}}(g)$ and $p_{\text{test}}(g)$ to represent the goal distribution during training and testing, respectively. *I* uses the same $p_{\text{train}}(g)$ and $p_{\text{test}}(g)$, *S* introduces spurious correlations in $p_{\text{train}}(g)$ but remove them in $p_{\text{test}}(g)$, and *C* contains more similar sub-goals in $p_{\text{test}}(g)$ than in $p_{\text{train}}(g)$. The details of these settings in are briefly summarized in the following (details in Appendix C.2.1):

- *Stack*: We design this manipulation task inspired by the CausalWorld [44], where the agent must stack objects to match specific shapes and colors. In *Stack-S*, we let the same shape have the same color in $p_{\text{train}}(g)$ but randomly sample the color and shape in $p_{\text{test}}(g)$. In *Stack-C*, the maximum number of object is two in $p_{\text{train}}(g)$ but five in $p_{\text{test}}(g)$.

- *Unlock*: We design this indoor house-holding task for the agent to collect a key to open doors. This environment is built upon the Minigrid [45]. In *Stack-S*, the door and the key are always in the same row in $p_{\text{train}}(g)$ but uniformly sample in $p_{\text{test}}(g)$. In *Unlock-C*, there are one door in $p_{\text{train}}(g)$ but two doors in $p_{\text{test}}(g)$.

- *Crash*: The occurrence of accidents usually relies on causality, e.g., an autonomous vehicle (AV) collides with a jaywalker because its view is blocked by another car [46]. We design such a crash scenario based on highway-env [47], where the goals are to create crashes between a pedestrian and different AVs. In *Stack-S*, the initial distance between AV and pedestrian is a constant in $p_{\text{train}}(g)$ but irrelevant in $p_{\text{test}}(g)$. In *Stack-C*, there is one pedestrian in $p_{\text{train}}(g)$ but two in $p_{\text{test}}(g)$.

- *Chemistry* [43]: There are 10 nodes with different colors. An underlying causal graph controls the color-changing mechanism of all nodes. In one step, the agent changes the color of one node. The goal is to match the given colors of all nodes. In the spuriousness setting, we let all nodes have the same target color. There is no composition setting in this environment.

We use the following methods as our baselines to fairly demonstrate the advantages of GRADER. **SAC:** [48] Soft Actor-Critic is a well-known model-free RL method that uses entropy to increase the diversity of action. **ICIN:** [25] It uses DAgger [49] to learn goal-conditioned policy with the causal graph estimated from the expert policy. We assume it can access the true causal graph for supervised learning. **PETS:** [50] We consider the ensemble transition model with random shoot planning as one baseline, which achieves generalization with the uncertainty-aware design. **TICSA:** [18] This is a causal-augmented MBRL method that simultaneously optimizes a soft adjacent matrix (representing the causality) and a transition model. **ICIL:** [16] This method proposes an invariant feature learning structure that captures the implicit causality of multiple tasks. We only use it for transition model learning since the original method is designed for imitation learning. **GNN:** [51] Since graph neural networks are good at learning structural information, we implement a GNN-based baseline using Relational Graph Convolutional Network.

## 4.2 Results Discussion

**Overall Performance (Q1)** We compare the testing reward of all methods under nine tasks and summarize the results in Table 1 to demonstrate the overall performance. Generally, our method outperforms baselines in all tasks except *Stack-I* because this task is too simple for all methods. We note that the gap between our method and baselines in *S* and *C* settings is more significant than in the *I* setting, showing that our method still works well in the non-trivial generalization task. As a

Table 1: Success rate (%) for nine settings in three environments. **Bold** font means the best.

| Method | Stack-I | Stack-S | Stack-C | Unlock-I | Unlock-S | Unlock-C | Crash-I | Crash-S | Crash-C |
|---|---|---|---|---|---|---|---|---|---|
| SAC | 34.7±16.1 | 22.1±14.0 | 31.7±5.1 | 0.1±0.5 | 0.0±0.2 | 0.4±1.7 | 22.5±17.6 | 18.6±8.7 | 6.7±3.8 |
| ICIN | 71.8±6.9 | 71.0±7.4 | 58.6±8.3 | 31.7±9.6 | 32.7±8.6 | 31.5±8.5 | 27.9±6.1 | 15.8±17.2 | 7.8±8.8 |
| PETS | **97.2±6.9** | 77.7±13.5 | 73.7±10.3 | 59.5±7.2 | 20.6±5.9 | 28.3±10.0 | 52.3±11.5 | 44.6±12.5 | 37.1±5.1 |
| TICSA | 85.9±8.4 | 88.8±10.1 | 76.2±8.3 | 58.5±12.3 | 33.6±14.3 | 29.8±8.3 | 68.9±5.9 | 56.8±8.6 | 15.0±8.2 |
| ICIL | 93.7±5.9 | 81.2±14.4 | 62.8±13.0 | **67.1±11.6** | 15.9±4.7 | 53.6±15.3 | 55.3±20.9 | 21.7±17.7 | 14.3±7.3 |
| GNN | 45.7±9.1 | 39.0±10.4 | 41.7±8.6 | 3.4±2.3 | 3.4±2.4 | 4.5±3.0 | 4.2±4.0 | 5.1±5.1 | 3.8±2.8 |
| Score | 92.7±7.4 | 90.5±7.5 | 73.9±8.5 | 44.9±28.1 | 23.1±7.6 | 36.2±30.1 | 42.3±17.5 | 53.4±18.7 | 8.4±6.1 |
| Full | 92.9±6.3 | 86.0±9.5 | 75.7±10.3 | 63.8±9.2 | 18.3±7.4 | 53.7±14.3 | 69.8±14.0 | 52.6±12.8 | 42.0±17.2 |
| Offline | 96.8±5.8 | 95.4±6.1 | 81.4±7.8 | 13.8±8.1 | 13.9±7.5 | 11.7±6.9 | 13.1±16.2 | 30.2±16.5 | 14.9±12.4 |
| GRADER | 95.6±5.4 | **97.6±6.0** | **93.7±8.4** | 64.2±9.1 | **61.4±4.4** | **82.1±9.2** | **91.5±4.4** | **84.3±10.0** | **84.7±7.3** |

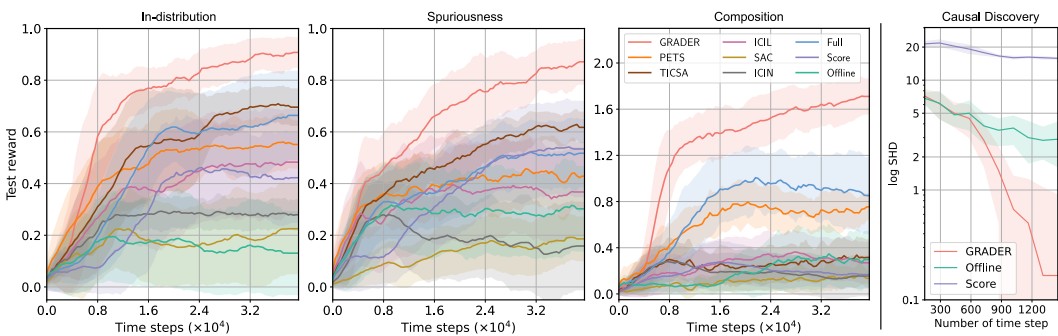

Figure 4: **Left:** Test reward of the *Crash* environment calculated with 30 trails. **Right:** The accuracy of causal graph discovery with samples from GRADER, Score, and Offline.

model-free method, SAC fails in all three tasks of *Unlock* and *Crash* environments since they have very sparse rewards. Without learning the causal structure of the environment, PETS even cannot fully solve *Unlock-S*, *Unlock-C*, and all *Crash* tasks. Both TICSA and ICIL learn the causality underlying the task so that they are relatively better than SAC and PETS. However, they are still worse than GRADER in two generalization settings because of the unstable and inefficient causal reasoning mechanism. We also find that even if ICIN is given the true causal graph, the policy learning part cannot efficiently leverage the causality, leading to worse performance in generalization settings.

To further analyze the tendency of learning, we plot the curves of all methods under *Crash* in Figure 4. Our method quickly learns to solve tasks at the beginning of the training, demonstrating high data efficiency. GRADER also outperforms other methods with large gaps in the later training phase. The training figures of the other two environments can be found in Appendix B.1.

**Importance of Policy Learning (Q2)** As we mentioned in Section 3.3, we empirically compare GRADER and **Offline** [52], which uses data collected from offline random policy, and plot results in the right part of Figure 4. We use SHD [53] to compute the distance between the estimated causal graph and the true causal graph. The true causal graph for each environment can be found in Appendix B.2. When we only use samples collected offline by random policy, we cannot obtain variables' values that require long-horizon reasoning, e.g., the door can be opened only if the agent is close to the door and has the key. As a consequence, the causal graph obtained by Offline harms the performance, as shown in Figure 4. Instead, GRADER gradually explores more regions and quickly obtains the true causal graph when we iteratively discover the causal graph and update the policy.

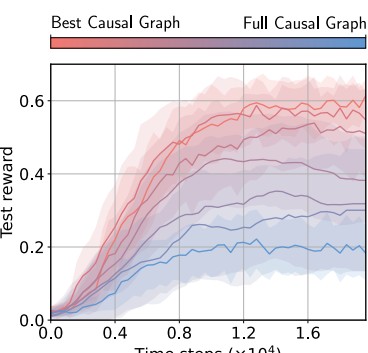

Figure 5: Influence of different causal graphs in *Unlock-S*.

**Advantage of Data-efficient Causal Discovery (Q3)** To show the advantage of proposed constraint-based methods, we design a model named **Score** that optimizes a soft adjacent matrix using score-based method [54], which is recently combined with NN for differentiable training, for example,

Table 2: Discovery results on *Chemistry* environment (GRADER / Score). **Bold** font means the best.

| Metric | Collider | Chain | Jungle | Full |
|--------|----------|-------|--------|------|
| SHD ($\downarrow$) | **3.70**$\pm$**1.79**/15.4$\pm$7.03 | **2.80**$\pm$**1.83**/14.0$\pm$1.18 | **7.00**$\pm$**2.19**/13.8$\pm$0.40 | **2.40**$\pm$**1.20**/11.0$\pm$5.31 |
| Accuracy ($\uparrow$) | **0.99**$\pm$**0.00**/0.87$\pm$0.06 | **0.99**$\pm$**0.00**/0.88$\pm$0.01 | **0.98**$\pm$**0.00**/0.89$\pm$0.00 | **0.99**$\pm$**0.00**/0.91$\pm$0.09 |
| Precision ($\uparrow$) | **0.90**$\pm$**0.05**/0.73$\pm$0.10 | **0.94**$\pm$**0.04**/0.79$\pm$0.03 | 0.86$\pm$0.04/**0.88**$\pm$**0.01** | 1.00$\pm$0.00/1.00$\pm$0.00 |
| Recall ($\uparrow$) | **0.99**$\pm$**0.02**/0.83$\pm$0.07 | **0.96**$\pm$**0.03**/0.73$\pm$0.00 | **0.96**$\pm$**0.02**/0.73$\pm$0.00 | **0.96**$\pm$**0.02**/0.83$\pm$0.17 |
| F-score ($\uparrow$) | **0.94**$\pm$**0.03**/0.77$\pm$0.06 | **0.95**$\pm$**0.03**/0.76$\pm$0.02 | **0.91**$\pm$**0.03**/0.80$\pm$0.00 | **0.98**$\pm$**0.01**/0.90$\pm$0.10 |

in TICSA. According to the discovery accuracy shown in the right part of Figure 4, we find that score-based discovery is inefficient. Based on the performance of the Score model summarized in Table 1, we also conclude that it is not as good as our constraint-based method and has a large variance due to the unstable learning of the causal graph.

**Influence of Causal Graph (Q4)** To illustrate the importance of the causal graph, we implement another variant of GRADER named **Full**, which uses a fixed full graph that connects all nodes between the sets $\mathcal{U}$ and $\mathcal{V}$. According to the performance shown in Table 1 and Figure 4, we find that the full graph achieves worse results than GRADER because of the redundant and spurious correlation. Intuitively, unrelated information causes additional noises to the learning procedure, and the spurious correlation creates a shortcut that makes the model extract wrong features, leading to worse results in the spuriousness generalization as shown in Table 1.

We then investigate how the correctness of the causal graph influences the performance. We use fixed graphs interpolating from the best causal graph to the full graph to train a GRADER model in *Unlock-S* and summarize the results in Figure 5. The more correct the graph is, the higher reward the agent obtains, which supports our statements in Section 3.4 that the causal graph is important for the reasoning tasks – a better causal graph helps the model have better task-solving performance.

## 4.3 Further Analysis of Causal Discovery

Finally, we conduct further analysis of the discovery performance on the *Chemistry* environment [43], which is a standard benchmark for evaluating causal discovery methods. In this environment, the colors of nodes are controlled by the causal graph, therefore, finding the true causal graph makes it much easier to achieve the goal that requires matching all target colors. The agent can discover the graph by doing interventions via interacting with the environment. We consider four types of causal graphs (*Collider*, *Chain*, *Jungle*, *Full*) with 10 nodes in this experiment.

The discovery performance is shown in Table 2 with five metrics indicating the classification error. We can see that GRADER outperforms the Score method in all 4 types of graphs. In Figure 6, we show the discovered graphs from GRADER (averaged over 10 seeds) and the true causal graphs of *Collider* and *Jungle* settings.

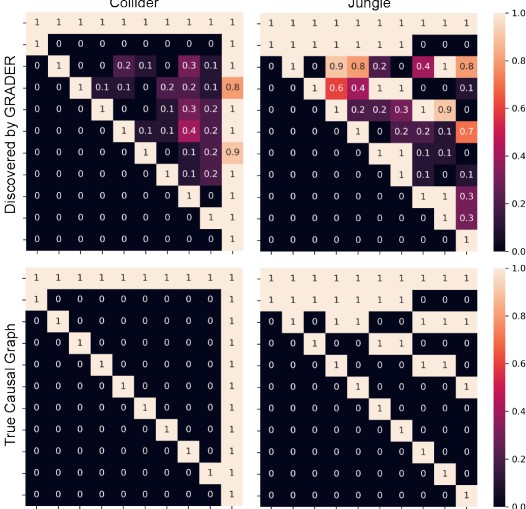

Figure 6: Top: Discovered causal graph from GRADER. Bottom: true causal graph.

We also show that GRADER achieves advantages over other baselines in solving the color-matching downstream task, which can be found with detailed experiment results in Appendix B.4.

## 5 Related Works

**RL Generalization** From the agent's view, algorithms focus on actively obtaining the structure of the task or environment. Some decompose the given task into sub-tasks [55, 56, 57, 5] and when they encounter an unseen task, they rearrange those sub-tasks to solve new tasks. Instead of

dividing tasks, Symbolic RL learns a program-based policy consisting of domain language [6, 8] or grammar [7, 58]. The generated program is then executed [9] to interact with the environment, which has the potential to solve unseen tasks by searching the symbolic space. From the other view, we can generate environments to augment agents' experience for better generalization. One straightforward way is data augmentation of image-based observation [59, 60, 61, 62, 63, 64]. When extended to other factors of the environment, *Domain Randomization* [65] and its variants [66, 67] are proposed. Considering the interaction between agent and environment, *Curriculum Learning* [68] also gradually generates difficult tasks to train generalizable agents.

**Goal-Conditioned RL (GCRL)** The generalization problem is naturally related to GCRL [26], which aims to train an agent for multiple tasks. From the optimization perspective, Universal Value Function [69], reward shaping [70], and latent dynamic model [71] are widely used tools to solve GCRL problem. Sub-goal generation [72] is another intuitive idea to tackle the long horizon with sparse reward, where the core thing is to make sure the generated sub-goals are solvable. Finally, *Hindsight Experience Replay (HER)* [34], belonging to the relabelling category, is a ground-breaking yet straightforward method that treats visited states as "fake" goals when the goal and state share the same space. Later on, improved versions of HER [73, 74, 75] were widely studied. One limitation is that we cannot directly use a visited state as a goal if the goal has pre-conditions. Similar to our setting, [76] and [77] convert the GCRL problem to variational inference by regarding control as inference [27]. [76] propose an EM framework under the HER setting and [77] treats the last state as the goal and estimates a shaped reward during training.

**RL with Causal Reasoning** Causality is now frequently discussed in the machine learning field to complement the interpretability of neural networks [29]. RL algorithms also incorporate causality to improve the reasoning capability [78]. For instance, [25] and [19] explicitly estimate causal structures with the interventional data obtained from the environment. These structures can be used to constraint output space [79] or adjust the buffer priority [20]. Building dynamic models in model-based RL [18, 80, 52] based on causal graphs is also widely studied recently. Implicitly, we can abstract the causal structure and formulate it using the *Block MDP* [12] setting or training multiple encoders to extract different kinds of representations [15]. Following the idea of invariant risk minimization [81], they assume task-relevant features are invariant and shared across all environments, which can be used as the only cause of the reward.

**Causal Discovery** Causal discovery [82] is a long-stand topic in economics and sociology, where the traditional methods can be generally categorized into constraint-based and score-based. Constraint-based methods [30] start from a complete graph and iteratively remove edges with conditional independent test [83, 84] as constraints. Score-based methods [39, 85] use metrics such as *Bayesian Information Criterion* [86] as scores and prefer edges that maximize the score given the dataset. Recently, researchers extend score-based methods with RL [87] or differentiable discovery [54, 88, 89]. The former selects edges with a learned policy, and the latter learns a soft adjacent matrix with observational or interventional data. Active intervention methods are also explored [90] to increase the efficiency of data collection and decrease the cost of conducting intervention [91].

## 6  Conclusion

This paper proposes a latent variable model that injects a causal graph reasoning process into transition model learning and planning to solve GCRL problems under the generalization setting. We theoretically prove that our iterative optimization process can obtain the true causal graph. To evaluate the performance of the proposed method, we designed nine tasks in three environments. The comprehensive experiment results show that our method has better data efficiency and performance than baselines. Our method also provides interpretability by the explicitly discovered causal graph.

The main limitation of this work is that the explicit estimation of causal structure does not scale well to the number of nodes. Developing efficient gradient-based discovery methods could be a promising direction. In addition, the factorized state and action space assumption may restrict the usage of this work to semantic representations, which need to be processed with abstraction methods. We further discuss the potential negative social impact and additional limitations in Appendix C.3.

**Acknowledgements.**  We gratefully acknowledge support from the National Science Foundation under grant CAREER CNS-2047454.

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
