# OpenReview forum: "Generalizing Goal-Conditioned Reinforcement Learning with Variational Causal Reasoning"
_NeurIPS.cc/2022/Conference — NeurIPS 2022 Accept_

### Official Review · Reviewer_qSKr · 2022-07-08

**Rating:** 6
**Confidence:** 5
**Soundness:** 3 good
**Presentation:** 4 excellent
**Contribution:** 2 fair

**Summary:**

The paper proposes to augment Goal-Conditioned RL (GCRL) with Causal Graph (CG), and treats the causal graph as a latent variable.
The paper then proposes to use  interventional data to estimate the posterior over the underlying causal graph, and then proposes to use the causal graph to learn factorized action conditioned model and agent's policy.


=====

After rebuttal: I've read the rebuttal, and I remain by my rating.

**Questions:**

**Model learning**

- The authors proposes to represent the transition model, with an ensemble of recurrent models, such that they can take into account variable number of "causal parents" for each node of the causal graph. Such a design choice is commonly used in causality papers (already cited by authors [1, 2], where the authors learn structural and functional parameters. Structural parameters corresponds to a soft-adjacency matrix, and functional parameters correspond to a set of MLPs which take in input a set of causal parents. This design choice has also been used as an inductive bias for learning architectures which consists of ensemble of recurrent models, such that each model can decide which other models to pay attention to via the use of attention [3, 4].

[1] Differentiable causal discovery from interventional data, https://arxiv.org/abs/2007.01754  \
[2] Learning neural causal models from unknown interventions, https://arxiv.org/abs/1910.01075 \
[3] Neural Production Systems, https://arxiv.org/abs/2103.01937  \
[4] Recurrent Independent Mechanisms, https://arxiv.org/abs/1909.10893

**Chosen Method for  learning causal graph**

It may be useful to employ gradient based methods of learning causal graph as compared to methods based on independence testing. Recently, few methods have been proposed which allows inferring posterior over the learned causal graph [1, 2, 3].

[1] DiBS: Differentiable Bayesian Structure Learning, https://arxiv.org/abs/2105.11839 \
[2] BCDNets, https://proceedings.neurips.cc/paper/2021/hash/39799c18791e8d7eb29704fc5bc04ac8-Abstract.html \
[3] Bayesian Structure Learning with Generative Flow Networks, https://arxiv.org/abs/2202.13903


**RL benchmarks for causal reasoning**

It may be useful to evaluate the proposed method on set of problems which are specifically tailored towards measuring generalization in RL (in causal sense) [1]. It may also be useful to evaluate an "attention" (or GNN) based baseline which learns interactions between different causal variables to predict the future state for planning.

[1] Systematic Evaluation of Causal Discovery in Visual Model Based Reinforcement Learning, https://arxiv.org/abs/2107.00848



**Limitations:**

**Limitations**

The paper discusses two limitations: (a) the method may not scale well to large number of nodes, (b) assuming factorized state and action space restricts the scope to scenarios where only observables are low level information i.e., pixels in the case of image data. It may also be useful to discuss that **independent testing** i.e., the method used to learn causal graph may not scale well as compared to gradient based approaches of learning causal graph.




**Strengths And Weaknesses:**

**Clarify**

- The paper is easy to read and well written.

**Originality**

- The individual elements are known in causality literature, and various other works (most of them are cited by the authors). The contribution lies in using these ideas (for ex. planning with a factorial action conditioned model) in the context of goal conditioned policies.

**Significance**

- The reviewer like the experiments where the authors focussed on using the transition models for doing planning. The reviewer believes it's a step in the right direction.

**Quality**

- The reviewer likes the experimental setup as it's tailored to test generalization (robustness to spurious correlations and compositional generalization).

---

> ### Author Response · Authors · 2022-08-02
> **Response to Reviewer qSKr**
>
> We thank the reviewer for valuable feedback and acknowledgment of our experiment design. We address the reviewer’s concerns in the following.
>
> ### **Q1: Causal transition model design.**
> > This design choice of causal-based transition model has also been used as an inductive bias for learning architectures which consists of an ensemble of recurrent models.
>
> We thank the reviewer for providing more literature on structural causal model design. This is definitely an important direction to capture more accurate causal relations. We add more discussion about these model design methods in the revised manuscript.
>
>
> ### **Q2: Gradient-based causal discovery.**
> > It may be useful to employ gradient-based methods of learning causal graphs as compared to methods based on independence testing. Some recent methods allow inferring posterior over the learned causal graph.
>
> We thank the reviewer for providing more literature on causal structure learning. These posterior estimation methods are useful in our latent variable model setting. We add more discussion about the gradient-based methods in the revised manuscript.
>
> The gradient-based discovery method are widely investigated recently for large datasets since they have good scalability. However, these methods also require lots of training data to converge. In Online RL, we don’t have enough data at the beginning of training. Thus, constraint-based methods are more suitable for causal RL tasks. Empirically, we compared with the score-based discovery methods (named Score) implemented based on [2] in our experiment, which does not show good performance with limited data samples.
>
> Although our constraint-based causal discovery does not scale as well as the score-based methods, our proposed independence tests achieve a time complexity of $\Omega(|S|(|S|+|A|))$, which is tolerable for most RL problems with lower dimensional state space. Empirical studies also show that our independent tests enjoy better data efficiency.
>
>
>
>
> ### **Q3: Additional experiments on new benchmark.**
> > Evaluate the proposed method on a set of problems that are specifically tailored toward measuring generalization in RL (in a causal sense) [1].
>
> We conduct additional discovery and RL experiments using the Chemistry environment in [1] as suggested by the reviewer. The results are provided in the general response to all reviewers.
>
> We have two main conclusions:
> (1) Our GRADER captures the true underlying causal graph with very high data efficiency.
> (2) Our GRADER outperforms all other baselines in the downstream RL task in both ID and OOD settings.
>
>
>
> ### **Q4: Add a GNN baseline on ALL benchmarks.**
> > Evaluate an "attention" (or GNN) based baseline which learns interactions between different causal variables to predict the future state for planning.
>
> We add a GNN baseline in all environments including Stack, Unlock, Crash, and Chemistry [1]. The results are shown in the general response to all reviewers.
> The conclusion is that the GNN method has poor performance in most of the settings since the causality cannot be easily discovered with GNN learning algorithms.
>
>
> ### **Q5: Scalability of independence test.**
> > It may also be useful to discuss that independent testing i.e., the method used to learn causal graphs may not scale well as compared to gradient-based approaches to learning causal graphs.
>
> Thanks for pointing out this potential limitation. We add this limitation in Section 6 and Appendix.C.4 of the revised manuscript. As discussed in question Q2, there may exist a trade-off between scalability and efficiency. Developing a scalable and efficient discovery method is definitely an important future direction.
>
> ---
>
> [1] [Ke, Nan Rosemary, et al. "Systematic evaluation of causal discovery in visual model-based reinforcement learning." arXiv preprint arXiv:2107.00848 (2021)](https://arxiv.org/abs/2107.00848)
>
> [2] [Brouillard, Philippe, et al. "Differentiable causal discovery from interventional data." Advances in Neural Information Processing Systems 33 (2020): 21865-21877](https://papers.nips.cc/paper/2020/hash/f8b7aa3a0d349d9562b424160ad18612-Abstract.html)

---

> ### Author Response · Authors · 2022-08-08
> **One day left! Did the response address your concerns?**
>
> Dear Reviewer qSKr,
>
> Thanks for asking questions and providing so many related works to improve this paper. We provide additional experiment results on the Chemistry benchmark as you requested. We kindly ask the reviewer to reassess the paper in light of our response. If all concerns are addressed, please consider raising the score. If not, we are very glad to have further discussions.
>
> Best,
>
> Authors of Paper7437

---

### Official Review · Reviewer_7kq2 · 2022-07-10

**Rating:** 6
**Confidence:** 4
**Soundness:** 3 good
**Presentation:** 3 good
**Contribution:** 3 good

**Summary:**

Paper investigates the problem of model-based RL or planning. It proposes to decompose the transition model into a causal graph and some parameterized functions. The causal discovery here is formulated as learning a latent variable model using VI, where the causal graph is the latent variable. Then the transition loss is used to find the best set of parameters. Experiments on several synthetic domains including one designated for causality show some promises over state-of-the-art model-based RL avenues.

**Questions:**

See [Weaknesses]


**Limitations:**

The authors mentioned that their limitation could be the inability of scaling to causal graphs with more nodes. However, I also doubt whether this approach could be applied to real-world domains with non-symbolic input, ex. pixel-based or other raw sensory input. Extracting symbolic input from these sources could impose new challenges on observation uncertainty, which cold be pivotal to causal discovery and model learning.

**Strengths And Weaknesses:**

[Stengths]

+Paper is overall clear and well-written. The authors also provide the code and indeed it helps with understanding the idea here.

+I found the formulation intuitive and reasonable. The math on identifiability and performance guarantee could be a bit dense but I didn't find any major issue.

+The advantages over MBRL baselines are impressive, especially in the challenging car crash domain. With the code already there, I'm sure the results are reproducible if the authors can release it at some point.

[Weaknesses]

I don't have major concerns at this point. But I will appreciate it if the following questions can be addressed in a rebuttal.

-The authors claim that their approach is for "goal-based RL". But it seems that goal is not necessary for all their formulations and implementations. Why not just call it model-based RL or planning in general? Some clarifications here are needed.

-The causal discovery part might be clear in the formulation but the implementation details are yet to be unveiled. I also check the code, but as far as I could find the causal graphs are **pre-defined**. The authors are encouraged to elaborate more on this and some pointers to the code will be helpful.

---

> ### Author Response · Authors · 2022-08-02
> **Reponse to Reviewer 7kq2**
>
> We thank the reviewer for providing valuable feedback and finding our paper's formulation reasonable and experiment results impressive. We address the reviewer's concerns in the following.
>
> ### **Q1: Extension to general MBRL.**
> > The authors claim that their approach is for "goal-based RL". But it seems that goal is not necessary for all their formulations and implementations. Why not just call it model-based RL or planning in general? Some clarifications here are needed.
>
> The proposed method can be naturally extended to general model-based RL tasks since we learn a general causal transition model. However, in this paper, we investigate how causality helps (spurious and compositional) **generalization**. For that reason, we use the goal-conditioned formulation since we can create these two types of generalization by designing different goals.
>
> Another reason is that goal-conditioned tasks are usually challenging due to the **sparsity** of the reward in a long horizon (e.g.., only receiving a non-zero reward after achieving the goal), which is non-trivial as shown in recent works [1][2]. Our proposed method shows that causality helps solve this challenging task more efficiently.
>
>
> ### **Q2: Details of causal discovery.**
> > The implementation details about causal discovery are to be unveiled. I also check the code, but as far as I could find the causal graphs are pre-defined. The authors are encouraged to elaborate more on this and some pointers to the code will be helpful.
>
> Due to the space limitation, we only provide the sketch of causal discovery and ignore all details. We add implementation details and an algorithm summary about the conditional independent test in **Appendix.C.1**.
>
> The discovery algorithm is implemented in script `/code/grader/discover.py` and the function is called in `/code/grader/grader.py` Line 66. The discovered causal graph is then used in `/code/grader/planner.py` Line 354. In the code provided in the supplementary, instead of using the discovered causal graph, we provide pre-defined causal graphs to check the best performance of our model. We have updated the code that uses discovered causal graph during training. Please check the updated supplementary.
>
>
> ### **Q3: Extension to raw sensory input.**
> > Could this approach be applied to real-world domains with non-symbolic input, ex. pixel-based or other raw sensory input?
>
> We acknowledge that current algorithms **cannot** be directly used for raw image input since we assume the action and state spaces are abstracted to object-level or event-level representations. However, there are already some methods [3][4] that investigate how to abstract symbolic variables from raw sensory input in reinforcement learning or use latent causal dynamics [6] to conduct independent component analysis from image observations. Combined with these methods, our approach can be extended to broader usage.
>
>
> ---
>
> [1] [Hansen-Estruch, Philippe, et al. "Bisimulation Makes Analogies in Goal-Conditioned Reinforcement Learning." arXiv preprint arXiv:2204.13060 (2022)](https://arxiv.org/pdf/2204.13060.pdf)
>
> [2] [Li, Yunfei, et al. "Phasic Self-Imitative Reduction for Sparse-Reward Goal-Conditioned Reinforcement Learning." International Conference on Machine Learning. PMLR, 2022](https://arxiv.org/abs/2206.12030)
>
> [3] [Abel, David. "A theory of abstraction in reinforcement learning." arXiv preprint arXiv:2203.00397 (2022)](https://arxiv.org/abs/2203.00397)
>
> [4] [Shanahan, Murray, and Melanie Mitchell. "Abstraction for Deep Reinforcement Learning." arXiv preprint arXiv:2202.05839 (2022)](https://arxiv.org/abs/2202.05839)
>
> [5] [Yao, Weiran, Guangyi Chen, and Kun Zhang. "Learning Latent Causal Dynamics." arXiv preprint arXiv:2202.04828 (2022)](https://arxiv.org/abs/2202.04828)

---

> > ### Comment · Reviewer_7kq2 · 2022-08-09
> > **Responses to author's rebuttal**
> >
> > I would like to thank the authors for their timely responses. My concerns on pre-defined causal graphs and limitations have been basically addressed. However, the comments on MBRL vs goal-based RL still stand. The authors mentioned that they choose to be allied with the latter due to the flexibility of designing generalization tests. If my understanding is correct, MBRL works well with goal-based task specifications. The authors did mention the following in the paper:
> > ```
> > The formulation we derived so far is highly correlated to the model-based RL framework [37]. However, the main difference is that we obtain it with variational inference by regarding the causal graph as a latent variable
> > ```
> > To get it straight: if you're learning and planning with a model and reward function, you're doing model-based planning but not goal-based RL, no matter how you obtain the model. I'm totally fine with a paper that contributes a novel and sound avenue to model learning in MBRL. This might not be a big issue at this point but I do encourage the authors to rethink the position of their approach.

---

> > > ### Author Response · Authors · 2022-08-09
> > > **Thanks for your response!**
> > >
> > > We are glad that our response addressed most of your concerns.
> > > As for the comments on MBRL v.s GCRL, we will carefully think about the position of our method to avoid ambiguity.
> > > Since MBRL is acknowledged as a powerful solution to the GCRL problem [1], we believe the proposed causal reasoning frameworks could lead to some extensive works in both fields.
> > > In this paper, GRADER aims to solve the goal-conditioned problem with a similar optimization framework as MBRL (including its model learning and planning).
> > >
> > > [1] [Liu, Minghuan, Menghui Zhu, and Weinan Zhang. "Goal-conditioned reinforcement learning: Problems and solutions.", 2022](https://arxiv.org/pdf/2201.08299.pdf)

---

> ### Author Response · Authors · 2022-08-08
> **One day left! Did the response address your concerns?**
>
> Dear Reviewer 7kq2,
>
> Thanks for asking questions about the clarification and goal-conditioned RL setting. We provide detailed responses to your questions. We kindly ask the reviewer to reassess the paper in light of our response. If all concerns are addressed, please consider raising the score. If not, we are very glad to have further discussions.
>
> Best,
>
> Authors of Paper7437

---

### Official Review · Reviewer_zrjV · 2022-07-12

**Rating:** 6
**Confidence:** 4
**Soundness:** 2 fair
**Presentation:** 3 good
**Contribution:** 2 fair

**Summary:**

Summary: the paper proposes an algorithm for learning causl strutures from the environment and using it to act and plan. The papers presents experiments on OOD generalization on either spurious correlations or compositional generalization.

**Questions:**

- It's unclear if the underlying causal structure of the environment for the stacking and door opening experiments are "interesting", in the sense that changing the color or backgrounds of the object has no causal influence on anything else in the env.

- Can't the spurious correlation experiment be solved by only focusing on learning what is a causal variables (compared to learning the causal mechanisms).



**Limitations:**

One of the major limitations of this paper is that the theoretical results the paper presents is based on somewhat unrealistic assumptions. However, this is not discussed properly in the paper.

 One of the main claims of the paper is the identifiability proof in that the algorithm can always identify the true graph. However, this proof is based on assumptions that generally may not hold true, espiecally in high-dimensional data. Another major reason why causal identifiability is so challenging in low-level data is that the causal variables are often latent, making the interventions unknown in the latent space. this is not discussed in the paper.

However, the proof that the authors presents in the paper assumes 2 things, one being that the mapping from the low-level observational space to latent causal variables are trivial and can be easily obtained, and second, that the independence test has perfect information, in that it can tell perfectly which variable cases which other variable upon intervention.

I think both of these assumptions generally do not hold true. Discovering latent states is non-trivial, and Im not sure if faithfulness generally holds either.

Another point that the authors have not discussed is that the interventions are unknown. The intervention is only observed on low-level images, but it's unknown to the abstract (latent) causal variables. I do not see a discussion on this anywhere in the paper.

**Strengths And Weaknesses:**

Pros:

- Causal models hold promise to better generalization, especially in RL. Causal models focuses on the right causal variables as well as causal mechanisms between variables.  Hence, this paper addresses an very important problem, espeically in RL.

- The experimental results are interesting, especially the OOD generalization experiments.

Cons:

There are some limitations of the work that would be nice to be addressed.

- One of the main claims of the paper is the identifiability proof in that the algorithm can always identify the true graph. However, this proof is based on assumptions that generally may not hold true, espiecally in high-dimensional data. Another major reason why causal identifiability is so challenging in low-level data is that the causal variables are often latent, making the interventions unknown in the latent space. this is not discussed in the paper.

However, the proof that the authors presents in the paper assumes 2 things, one being that the mapping from the low-level observational space to latent causal variables are trivial and can be easily obtained, and second, that the independence test has perfect information, in that it can tell perfectly which variable cases which other variable upon intervention.

I think both of these assumptions generally do not hold true. Discovering latent states is non-trivial, and Im not sure if faithfulness generally holds either.

Another point that the authors have not discussed is that the interventions are unknown. The intervention is only observed on low-level images, but it's unknown to the abstract (latent) causal variables. I do not see a discussion on this anywhere in the paper.

- The experiments are interesting. However, the spurious correlation experiments can be solved without learning the underlying causal mechanisms. The model would only need to discover the causal variables (but not necessarily mechanisms). The compositional generalization tasks that the authors proposed is more interesting, but again, the agent doesnt  necessarily have to know the undelrying causal graph in order to generalize. For example, stacking 2 objects and stacking 5 objects as proposed by the authors. A model that has not learned the underlying causal structure of the environment might also be able to generalize to this.

- It's also unclear if the underlying causal structure of the environment for the stacking and door opening experiments are "interesting", in the sense that changing the color or backgrounds of the object has no causal influence on anything else in the env.

- In order to test to see if the model has indeed captured the underlying causal structure, I would propose that the authors use either

1. "Alchemy: A structured task distribution for meta-reinforcement learning"
2. "Systematic Evaluation of Causal Discovery in Visual Model Based Reinforcement Learning"

Both of these environments have more complicated and interesting causal graphs. There is also a set of interesting OOD experiments that can be performed on these experiments.

---

> ### Author Response · Authors · 2022-08-02
> **Response to Reviewer zrjV (1/2)**
>
> We thank the reviewer for valuable feedback and acknowledgment of our experiment design.
> We also want to emphasize that our core contribution lies in improving the generalization of decision-making agents using causality. Thorough empirical studies, as well as theoretical analysis under some common assumptions, support that our method could be a good solution for GCRL. We address the reviewer's concerns as follows and add discussions of limitations in our revised manuscript.
>
> ### **Q1: Theoretical results is based on somewhat unrealistic assumptions.**
> > The proof assumes 2 things, one being that the mapping from the low-level observational space to latent causal variables are trivial and can be easily obtained, and second, that the independence test has perfect information, in that it can tell perfectly which variable causes which other variable upon intervention.
>
> We thank the reviewers for pointing out the potential limitation in some parts of our theoretical proof. We add more discussion about the assumptions in Appendix.C.3 of the revised manuscript. The detailed responses are as follows.
>
> **1. Regarding "High-dimensional data abstraction is non-trivial".**
>
> * We might give a wrong impression to the reviewer that we work on high-dimensional data. The challenge the reviewer mentioned exists but it is an orthogonal problem to the problem that we are solving.
>
> * We acknowledge that high-dimensional data abstraction is another non-trivial problem, as investigated in recent literature [10][11]. However, even if the agent could access low-dimensional states, conducting efficient model learning to achieve generalization against spuriousness and compositionality is also non-trivial. So in this work, we focus on how to use causality to preclude unnecessary dependencies between states and actions to achieve better generalization in downstream decision-making tasks. Our experiments show that even with low-dimensional nodes, most existing approaches still have issues with reliably generalizing in the OOD setting.
>
> * There are many real-world applications (mentioned in our experiments) that do not necessarily use high-dimensional data as state observations, such as robot manipulation and robot navigation, where all the low-dimensional physical states are accessible by the sensors and are sufficient for the agent to achieve goals. We believe these tasks are already non-trivial (e.g., under long-horizon setting with sparse reward), and could have a wide range of applications.
>
> **2. Regarding "Independence test has perfect information".**
>
> * Faithfulness and Markov properties are commonly used in causal discovery literature such as [1][2]. It is claimed in [3] that the oracle independent test can be ensured by the satisfaction of Markov property and faithfulness. Recent work [4] also assumes the oracle of conditional independent test in its Assumption 2.1. Practically, the oracle conditional independence test can be implemented with certain sub-linear sample complexity, as is investigated in [5].
>
> * In reinforcement learning tasks, agents interact with the environment by doing interventions, which is achieved by assigning values to action nodes. Then, the intervention results are reflected by the states. Under fully observable Markov settings, the value of these states contains all information about the intervention. Thus, our RL setting usually satisfies the assumptions we use in the theoretical proof.
>
>
>
>
>
>
> ### **Q2: The authors have not discussed is that the interventions are unknown.**
> > The intervention is only observed on low-level images, but it's unknown to the abstract (latent) causal variables. Causal variables are often latent, making the interventions unknown in the latent space.
>
> We acknowledge that unknown intervention in the latent space is another important research area for causal discovery.
> However, in this work, we assume there is **no** unmeasured (latent) causal variable and our MDP is fully observable with abstraction.
> Different from learning latent causal variables to model the transition (e.g., in [6]), we directly use abstracted states and actions in MDP as causal variables. We only need to discover the causal relations, i.e. the edge connections and the parameters of structural causal models. Therefore, the target of intervention in our paper is **known** because the intervention is directly applied to action nodes (e.g. the agent accelerates or moves).
>
> In our experiments, we don't use low-level images as state observation. All the actions and states in our works are low-dimensional variables that have clear **semantic meanings**, e.g. colors, positions, speed, stacking, accelerating, picking, etc.

---

> > ### Author Response · Authors · 2022-08-02
> > **Response to Reviewer zrjV (2/2)**
> >
> > ### **Q3: Necessity of learning causal graph for generalization.**
> > > Learning the underlying causal graph is not necessary for the agent to generalize in the two experiment settings. The model would only need to discover the causal variables (but not necessarily mechanisms).
> >
> > We cannot agree with the reviewer on this point as it is clearly **against** the findings of our experiments and not supported by the theoretical analysis. In our method, the explicit learning of causal graphs is important to solve downstream RL tasks. The detailed reasons are as follows.
> >
> > 1. In Lemma 1, we show a better causal graph can lead to a better convergence of the transition model for factorized MDP. Intuitively, knowing causality improves the following two types of generalization: (1) In **spurious** generalization, an agent could get rid of spurious correlation to conduct planning that leads to the goal with accurate models, as we demonstrate in Lemma 2 and 3. (2) In **compositional** generalization, our designed tasks can be decomposed into similar sub-tasks so that the agent could reuse the learned causal graph to solve these tasks more efficiently.
> >
> > 2. Discovering the underlying causal graph **empirically** improves the generalization. Our experiment shows that even if the causal variables are known, the performance is still not good unless we identify the entire structure of the causal graph. In Table 1, the Full method also knows all causal variables but the gap between GRADER and Full is larger in two generalization settings than in the in-distribution setting. Also, in Figure 5, all models know the causal variables, but the performance is better when the graph is closer to the true graph.
> >
> > ### **Q4: Causal variables in experiment design.**
> > > It's also unclear if the underlying causal structure of the environment for the stacking and door opening experiments is "interesting", in the sense that changing the color or backgrounds of the object has no causal influence on anything else in the env.
> >
> > We humbly argue that color is an important factor that could cause **spuriousness**. For example, Colored MNIST [8] contains the spurious correlation between digits and color, and [9] uses color as a causal variable in the representation learning. In our Stack experiment, we intentionally create spuriousness between color and shape to test the **generalization** capability of all methods. Empirical results in our experiments show that baselines has a significant performance drop in two OOD generalization settings.
> >
> > We don't use backgrounds as variables in our experiment settings. However, backgrounds can also be used as a distractor that causes spurious correlation.
> >
> > ### **Q5: Additional experiments on new benchmark.**
> > > More environments for evaluating the underlying causal structure. There is also a set of interesting OOD experiments that can be performed on these experiments.
> >
> > We conduct additional discovery and RL experiments using the Chemistry environment in [7] as suggested by the reviewer. The results are provided in the general response to all reviewers.
> >
> > We have two main conclusions:
> > (1) GRADER captures the true underlying causal graph with very high data efficiency.
> > (2) GRADER outperforms all other baselines in the downstream RL task in both ID and OOD settings.
> >
> > ---
> >
> > [1] [Brouillard, Philippe, et al. "Differentiable causal discovery from interventional data." NeurIPS, 2020](https://papers.nips.cc/paper/2020/hash/f8b7aa3a0d349d9562b424160ad18612-Abstract.html)
> >
> > [2] [Chickering, David Maxwell. "Optimal structure identification with greedy search." JMLR, 2002](https://www.jmlr.org/papers/volume3/chickering02b/chickering02b.pdf)
> >
> > [3] [Addanki, Raghavendra, Andrew McGregor, and Cameron Musco. "Intervention efficient algorithms for approximate learning of causal graphs." ALT, 2021](https://arxiv.org/abs/2012.13976)
> >
> > [4] [Addanki, Raghavendra, et al. "Efficient intervention design for causal discovery with latents." ICML, 2020](https://arxiv.org/abs/2005.11736)
> >
> > [5] [Canonne, Clément L., et al. "Testing conditional independence of discrete distributions." (2018)](https://ieeexplore.ieee.org/document/8503255)
> >
> > [6] [Yao, Weiran, Guangyi Chen, and Kun Zhang. "Learning Latent Causal Dynamics." (2022)](https://arxiv.org/abs/2202.04828)
> >
> > [7] [Ke, Nan Rosemary, et al. "Systematic evaluation of causal discovery in visual model-based reinforcement learning." (2021)](https://arxiv.org/abs/2107.00848)
> >
> > [8] [Arjovsky, Martin, et al. "Invariant risk minimization." (2019)](https://arxiv.org/abs/1907.02893)
> >
> > [9] [Sontakke, Sumedh A., et al. "Causal curiosity: RL agents discovering self-supervised experiments for causal representation learning." ICML, 2021](https://arxiv.org/abs/2010.03110)
> >
> > [10] [Abel, David. "A theory of abstraction in reinforcement learning." (2022)](https://arxiv.org/abs/2203.00397)
> >
> > [11] [Shanahan, Murray, and Melanie Mitchell. "Abstraction for Deep Reinforcement Learning." (2022)](https://arxiv.org/abs/2202.05839)

---

> ### Author Response · Authors · 2022-08-08
> **One day left! Did the response address your concerns?**
>
> Dear Reviewer zrjV,
>
> Thanks for asking questions about the theoretical proof. We provide detailed responses to your questions and additional experiment results to show the advantages of our method. We kindly ask the reviewer to reassess the paper in light of our response. If all concerns are addressed, please consider raising the score. If not, we are very glad to have further discussions.
>
> Best,
>
> Authors of Paper7437

---

> ### Comment · Reviewer_zrjV · 2022-08-09
> **Response to authors**
>
> I'd like to thank the authors for their detailed feedback and for the hard  work done in taking the required comments into consideration. I appreciate the additional results on the chemistry environment and for addressing the limitations in terms of identifiability.
>
> One comment that I could like to clarify with is that I do fully agree that learning the causal structure (not just causal variables) are crucial for OOD generalization, my concern was mainly with the setup that the authors used in terms of experiments. To be specific, my concern was that the exp setup did not require the model to learn the structure, it only required the model to distinguish between causal and non-causal variables, but the results on the Chemistry environment resolved this concern, so now I am satisfied with the experiments).
>
> I am happy to raise the score.

---

> > ### Author Response · Authors · 2022-08-09
> > **Thanks for your response!**
> >
> > We are glad that our response addressed the reviewer's concern. We agree that the Chemistry environment improves the quality of the experiment part and we will add more discussion about the experiment results in the revised version.

---

### Author Response · Authors · 2022-08-02
**Additional Experiment Results (1/2)**

## **Experiments on the Chemistry environment [1]**

As suggested by reviewers zrjV and qSKr, we report the results of our methods and baselines on the Chemistry environment. We also add the results and discussion in Appendix C.2. The figure of the success rate curve during training is shown in this [**[Anonymous Link]**](https://ibb.co/X2B643V). The discovered causal graph is shown in this [**[Anonymous Link]**](https://ibb.co/SKpdFRc).

---------

### **Table 1: Causal discovery results (SHD)**
| Evaluation Metric      | SHD        |           | Accuracy   |           | Precision |           | Recall |           | F-score |         |
| :---------- | :--------: | :-------: | :--------: | :-------: | :-------: | :-------: | :----: | :-------: | :-----: | :-----: |
| **Method**  | **GRADER** | **Score** | **GRADER** | **Score** | **GRADER** | **Score** | **GRADER** | **Score** | **GRADER** | **Score** |
| Collider-10 | **3.70±1.79**  | 15.40±7.03 | **0.99±0.00** | 0.87±0.06 | **0.90±0.05**  | 0.73±0.10 | **0.99±0.02**  | 0.83±0.07 | **0.94±0.03**  | 0.77±0.06 |
| Chain-10    | **2.80±1.83**  | 14.00±1.18 | **0.99±0.00** | 0.88±0.01 | **0.94±0.04**  | 0.79±0.03 | **0.96±0.03**  | 0.73±0.00 | **0.95±0.03**  | 0.76±0.02 |
| Jungle-10   | **7.00±2.19**  | 13.80±0.40 | **0.98±0.00** | 0.89±0.00 | 0.86±0.04  | **0.88±0.01** | **0.96±0.02**  | 0.73±0.00 | **0.91±0.03**  | 0.80±0.00 |
| Full-10     | **2.40±1.20**  | 11.00±5.31 | **0.99±0.00** | 0.91±0.09 | 1.00±0.00  | 1.00±0.00 | **0.96±0.02**  | 0.83±0.17 | **0.98±0.01**  | 0.90±0.10 |

--------------

### **Table 2: RL downstream results of ID setting (Success rate)**
| Env / Method | SAC     | ICIN     | PETS     | TICSA    | ICIL     | GNN       | GRADER       | Score    | Full     |
| :----        | :-----: | :------: | :------: | :------: | :------: | :-------: | :----------: | :------: | :------: |
| Collider-10     | 0.0±0.0 | 29.8±7.2 | 0.6±0.8  | 70.1±4.2 | 1.3±1.3  | 7.3±4.3   | **85.5±3.4** | 53.0±4.1 | 70.3±5.3 |
| Chain-10        | 1.1±1.3 | 37.5±4.0 | 29.6±4.9 | 24.5±4.3 | 25.3±5.1 | 24.6±14.5 | **77.0±3.2** | 60.2±2.4 | 72.7±5.3 |
| Jungle-10       | 0.6±0.8 | 20.2±1.5 | 18.8±5.2 | 31.8±4.5 | 20.6±3.9 | 27.5±9.8  | **69.6±4.3** | 63.0±2.3 | 59.4±9.5 |
| Full-10         | 0.5±0.8 | 4.5±4.0  | 23.7±4.3 | 10.4±3.0 | 22.3±5.1 | 20.4±7.8  | **59.1±6.6** | 39.4±3.5 | 47.1±6.1 |

--------------

### **Table 3: RL downstream results of OOD setting (Success rate)**
| Env / Method | SAC     | ICIN     | PETS     | TICSA    | ICIL         | GNN      | GRADER       | Score    | Full    |
| :----        | :-----: | :------: | :------: | :------: | :----------: | :------: | :----------: | :------: | :-----: |
| Collider-5     | 0.0±0.0 | 53.3±1.6 | 87.2±8.5 | 96.6±1.4 | **97.0±2.0** | 72.8±7.5 | 95.8±2.6     | 92.4±3.5 | 87.8±4.4 |
| Chain-5        | 0.0±0.0 | 27.3±5.9 | 37.1±7.0 | 54.0±3.8 | 50.0±5.8     | 3.9±1.6  | **82.3±4.5** | 46.8±5.0 | 52.9±4.3 |
| Jungle-5       | 0.8±2.4 | 42.6±4.9 | 53.9±5.5 | 43.1±7.5 | 52.9±7.0     | 11.1±2.4 | **84.4±5.1** | 59.5±2.7 | 60.8±3.5 |
| Full-5         | 0.0±0.0 | 28.9±5.0 | 43.5±4.1 | 55.9±4.5 | 42.2±5.9     | 3.8±2.5  | **83.9±4.4** | 50.7±6.0 | 54.2±4.1 |

---

> ### Author Response · Authors · 2022-08-02
> **Additional Experiment Results (2/2)**
>
> ### **1. Experiment setting**
>
> * We use four types of causal graphs (collider, chain, jungle, full). The state is the color of all nodes. The action can change the color of one node at one time, then the colors of the remaining nodes are changed according to the causal graph. The downstream task is to change the color of nodes to match the target colors within maximum steps (T=10). A reward r=1 is received if all colors are matched. Results are reported with 200 episodes. We use planning horizon H=5.
>
> * We provide the causal discovery results in Table 1. All graphs have 10 nodes and 10 colors. Results are reported with 200 episodes.
>
> * We provide the RL downstream task results in Table 2 (ID setting) and Table 3 (OOD setting). The graphs have 10 nodes and 10 colors in the ID setting, and 5 nodes and 5 colors for the OOD setting.
>     In the ID setting, we randomly sample the target colors in the goal for both the training and testing stages.
>     While in the OOD setting, we set the target colors of all nodes to the same color to create spurious correlations during training, then randomly set the target colors during testing.
>
> ### **2. Results analysis.**
>
> * Our GRADER captures the true underlying causal graph with very high data efficiency compared to the score-based causal discovery methods. You may check the demonstration of the discovered causal graph in the [**[Anonymous Link]**](https://ibb.co/SKpdFRc).
>
> * Since the maximum step is limited, the tasks could be difficult to be fully solved. However, our GRADER outperforms all other baselines in all 4 settings. This advantage is attributed to the efficient causal discovery and model learning in GRADER.
>
> ### **3. Setting clarification**
>
> Besides, it's worth to mention some differences between the setting of the Chemistry environment [1] and our experiment.
>
> * [1] incorporates a causal graph to describe causal relations within the state space, while our causal graph describes state-action causal relations between all the actions, current states, and next states.
>
> * Our intervention can only be applied to action nodes, which is the common case in RL. This makes the problem challenging because it is hard to collect information by randomly doing interventions. In contrast, [1] assumes all states are controllable, which is easy for the agent to collect information by intervention.
>
> * Discovering the causal graph in this environment is simpler than our experiment settings since all causal variables can be intervened. In our settings, some cause and effect will not be observed until the agent achieves some progress in the task.
>
> ------
>
> ## **Add a GNN baseline in experiments**
>
> | Env / Method | SAC       | ICIN     | PETS         | TICSA    | ICIL         | GNN      | GRADER       | Score    | Full     | Offline  |
> | :----        | :-------: | :------: | :----------: | :------: | :----------: | :------: | :----------: | :------: | :------: | :------: |
> | Stack-I      | 34.7±16.1 | 71.8±6.9 | **97.2±6.9** | 85.9±8.4 | 93.7±5.9     | 45.7±9.1 | 95.6±5.4     | 92.7±7.4 | 92.9±6.3 | 96.8±5.8 |
> | Stack-S      | 22.1±14.0 | 71.0±7.4 | 77.7±13.5    | 88.8±10.1| 81.2±14.4    | 39.0±10.4| **97.6±6.0** | 90.5±7.5 | 86.0±9.5 | 95.4±6.1 |
> | Stack-C      | 31.7±5.1  | 58.6±8.3 | 73.7±10.3    | 76.2±8.3 | 62.8±13.0    | 41.7±8.6 | **93.7±8.4** | 73.9±8.5 | 75.7±10.3| 81.4±7.8 |
> | Unlock-I     | 0.1±0.5   | 31.7±9.6 | 59.5±7.2     | 58.5±12.3| **67.1±11.6**| 3.4±2.3  | 64.2±9.1     | 44.9±28.1| 63.8±9.2 | 13.8±8.1 |
> | Unlock-S     | 0.0±0.2   | 32.7±8.6 | 20.6±5.9     | 33.6±14.3| 15.9±4.7     | 3.4±2.4  | **61.4±4.4** | 23.1±7.6 | 18.3±7.4 | 13.9±7.5 |
> | Unlock-C     | 0.4±1.7   | 31.5±8.5 | 28.3±10.0    | 29.8±8.3 | 53.6±15.3    | 4.5±3.0  | **82.1±9.2** | 36.2±30.1| 53.7±14.3| 11.7±6.9 |
> | Driving-I    | 22.5±17.6 | 27.9±6.1 | 52.3±11.5    | 68.9±5.9 | 55.3±20.9    | 4.2±4.0  | **91.5±4.4** | 42.3±17.5| 69.8±14.0| 13.1±16.2|
> | Driving-S    | 18.6±8.7  | 15.8±17.2| 44.6±12.5    | 56.8±8.6 | 21.7±17.7    | 5.1±5.1  | **84.3±10.0**| 53.4±18.7| 52.6±12.8| 30.2±16.5|
> | Driving-C    | 6.7±3.8   | 7.8±8.8  | 37.1±5.1     | 15.0±8.2 | 14.3±7.3     | 3.8±2.8  | **84.7±7.3** | 8.4±6.1  | 42.0±17.2| 14.9±12.4|
>
> ### **1. Experiment setting.**
>
> As suggested by reviewer qSKr, we implement a GNN baseline using a 3-layer Relational Graph Convolutional Network (RGCN) [2]. All other hyper-parameters are the same as our GRADER. The results show that GNN fails to learn the accurate world models and could not generalize as well as our proposed GRADER in all environments. We add these results in Table.1 of the revised manuscript.
>
> ---
>
> [1] [Ke, Nan Rosemary, et al. "Systematic evaluation of causal discovery in visual model-based reinforcement learning.", 2021](https://arxiv.org/abs/2107.00848)
>
> [2] [Schlichtkrull, Michael, et al. "Modeling relational data with graph convolutional networks." European semantic web conference. Springer, Cham, 2018.](https://arxiv.org/abs/1703.06103)

---

### Author Response · Authors · 2022-08-05
**Reminder for Further Discussions**

Dear all reviewers,

Thanks again for your valuable suggestions and questions. We hope our new empirical results and discussions about theoretical proof as well as limitations have addressed the concerns raised in the first-round review. Since we have a limited discussion window, please feel free to post your response if you have further questions.

Best,

Authors of Paper7437

----------------

---

### Meta-Review · Area_Chair_ZT1H · 2022-08-31

**Recommendation:** Accept
**Confidence:** Less certain

**Metareview:**

Promising direction for incorporating casual reasoning and RL   although the scope of theory and experiments seems limited.

**Award:**

No

---

### Decision · Program_Chairs · 2022-09-14

Accept